

# The effects of biotic stress on the sexual reproduction process of flowering plants

Zhenzhen Li[1], Shuo Wang[1], Yike Wang[1], Hongxia Zhang[1], Lu Liu[1], Shiwen Su[2] and Sue Lin[1,3]

[1] College of Life and Environmental Science, Wenzhou University, Wenzhou, China
[2] Southern Zhejiang Key Laboratory of Crop Breeding, Wenzhou Vocational College of Science and Technology, Wenzhou, China
[3] Zhejiang Provincial Key Laboratory of Water Ecological Environment Treatment and Resource Protection, Wenzhou University, Wenzhou, China

## ABSTRACT

The sexual reproduction phase of flowering plants encompasses a multitude of physiological processes, including floral induction, floral organ morphogenesis, fertilization, and the maturation of seeds and fruits. In addition to being vital to the successful reproduction of the plants, these processes are also crucial to their adaptation to diverse environmental conditions. However, this phase is extremely complex and vulnerable to environmental impacts and constraints, with both biotic and abiotic stresses playing a significant role. Accumulating evidence has demonstrated that environmental stress has multifaceted impacts on plant sexual reproduction, leading to substantial losses in seed production and crop yield. Although several excellent reviews have explored the regulatory mechanisms of abiotic stresses (such as light and temperature stress) on the plant sexual reproduction process, particularly flowering time and gametophyte development, a comprehensive overview of the effects of biotic stresses is still lacking. Rather than comprehensively reviewing the massive amount of literature in this field, our review aims to leverage case studies to cover a wide range of mechanisms by which biotic stressors, including fungi, bacteria, viruses, parasitic plants, and herbivorous animals, affect the sexual reproduction process of flowering plants.

## INTRODUCTION

Flowering plants undergo a series of distinct stages throughout their life cycle, encompassing embryonic development, vegetative growth, reproductive development, and senescence. The reproductive development phase is critical to the success of the next generation (*Alvarez-Buylla et al., 2010*). The floral transition, which marks the shift from the vegetative phase to the reproductive phase, represents the most dramatic phase change in plant development. For this transition to occur, endogenous cues (such as age and phytohormones) need to integrate with environmental cues (such as light and low temperature) to coordinate this developmental switch (*Benaouda et al., 2023*; *Pagnussat & Gomez-Casati, 2024*). Once the decision to flower has been made, the shoot apical

Corresponding authors
Shiwen Su, sushiwen@wzvcst.edu.cn
Sue Lin, iamkari@163.com

meristem (SAM) undergoes volumetric enlargement and accelerates its division rate, progressively developing into the floral meristem (FM) (*Cucinotta et al., 2021*). Subsequently, the stem cell populations retained within the FM maintain a delicate balance between proliferation and differentiation rates, and this is crucial for ensuring the correct morphogenesis of individual floral organs situated at the apex of the pedicel and the elongation of the floral axis (*Sun et al., 2009*; *Lee et al., 2019*). This stage involves the differentiation of the floral organ primordia into five concentric whorls: the outermost whorl (the sepals), which is followed by the petals, stamens, carpels, and finally the ovules. The molecular mechanisms behind the formation of floral organs have been explained by the ABCDE model (*Irish, 2017*). The final stage is when the flower matures, accompanied by the release of pollen from the anther onto the stigma. The pollen germinates on the stigma, and the pollen tube then penetrates the ovules, resulting in the ultimate fusion of gametes to complete the double fertilization (*Edlund, Swanson & Preuss, 2004*; *Chapman & Goring, 2010*; *Meng et al., 2023*). After anthesis, the plants focus their energy on fruit development and seed formation (*Zhao et al., 2023*).

Several comprehensive reviews have explored the genetic and biochemical mechanisms governing plant reproduction, with a focus on the genes and epigenetic machinery involved in flowering induction, floral organ formation, gametophyte development, fertilization, and seed development (*Rieu et al., 2023*; *Takagi, Hempton & Imaizumi, 2023*; *Chow & Mosher, 2023*; *Wang et al., 2023*; *Zhang & Elomaa, 2024*). Plant reproduction is regulated not only by a complex intrinsic genetic network but also by diverse environmental cues, including both biotic and abiotic stresses (*De Storme & Geelen, 2014*; *Nawaz et al., 2023*; *Begcy, Mendes & De Storme, 2024*). The roles of abiotic stresses in regulating reproductive development have been well-documented in several excellent reviews (*Resentini et al., 2023*; *Patra et al., 2024*; *Begcy, Mendes & De Storme, 2024*; *Ye et al., 2024*; *Qian et al., 2025*). Biotic stresses, such as pathogens and pests, can regulate flowering time by altering the expression of key flowering genes (*e.g.*, FLC, FT, GI), disrupt floral organogenesis (*e.g.*, stigma, filament, anther), and impair pollen viability (*Tang et al., 2013*; *Lyons et al., 2015*; *Otulak, Kozieł & Garbaczewska, 2016*; *Rasmann et al., 2018*; *Fan et al., 2020*). However, in contrast to the extensive research on abiotic stresses, there is a notable scarcity of comprehensive reviews that delve into the effects of biotic stresses on plant reproductive development. The limited existing reviews, while relevant, tend to focus narrowly on the timing of floral induction, offer only a general overview of stress impacts across all stages of plant growth and development, or broadly cover plant strategies against both biotic and abiotic stressors, rather than providing a comprehensive analysis of the specific effects on the sexual reproduction process (*Nawaz et al., 2023*; *Patra et al., 2024*). To address this gap, this review focuses on summarizing the effects of biotic stress on the sexual reproduction of flowering plants (Table 1) and elucidates the underlying mechanisms involved by leveraging case studies. The review should appeal to researchers in the fields of biology and agronomy, with a particular resonance for those studying plant reproduction and environmental stress.

**Table 1 Summary of the impact of biotic stresses on the sexual reproductive processes of flowering plants discussed in this review.**

| Biological stress type | Stress source | Host plant(s) | Infection/ Feeding site | Impact/Damage | References |
|---|---|---|---|---|---|
| Fungi | *Peronospora parasitica* | Arabidopsis | Leaves | Accelerates flowering | *Korves & Bergelson (2003)* |
| | *Fusarium oxysporum* | Arabidopsis | Roots | Accelerates flowering | *Lyons et al. (2015)* |
| | *Piriformospora indica* | Arabidopsis and *Coleus forskohlii* | Roots | Accelerates flowering | *Kumari et al. (2003), Das et al. (2012)* |
| | *Pochonia chlamydosporia* | Arabidopsis and tomato (*Solanum lycopersicum*) | Roots | Accelerates flowering and improves seed yield | *Zavala-Gonzalez et al. (2015, 2017)* |
| | *Ustilaginoidea virens* | Rice (*Oryza sativa*) | Floral organs | Halts flowering and prevents fertilization | *Fan et al. (2015)* |
| | *Claviceps purpurea* | *Secale cereale*, barley (*Hordeum vulgare*), and wheat (*Triticum aestivum*) | Pistil | Impedes seed development | *Miedaner & Geiger (2015), Tente et al. (2021)* |
| | *Fusarium graminearum* | Maize (*Zea mays*) and wheat | Floral organs | Reduces yield | *Boenisch & Schäfer (2011)* |
| | *Ustilago tritici* | Wheat | Floral organs | Impedes seed development | *Thambugala et al. (2020)* |
| | *Sporisorium reilianum* | Maize | Floral organs | Reduces yield | *Wang et al. (2024)* |
| Viruses | *Prunus necrotic ringspot virus* | Apricot (*Prunus armeniaca*) | Pollen | Reduces pollen germination rate and slows pollen tube growth | *Amari et al. (2007)* |
| | *Tomato brown rugose fruit virus* | Tomato | Leaves and Floral organs | Reduces pollen germination rate | *Avni et al. (2022)* |
| | *Tobacco rattle virus* | Tobacco (*Nicotiana tabacum*) and pepper (*Capsicum annuum*) | Anthers | Reduces floral/pollen number and induces pollen degeneration | *Otulak, Kozieł & Garbaczewska (2016)* |
| | *Raspberry bushy dwarf virus* | *Torenia fournieri* | Stigma | Inhibits fertilization | *Isogai et al. (2014)* |
| | *Zucchini yellow mosaic virus* | Wild squash (*Cucurbita pepo* subsp. *Texana*) | Leaves | Reduces yield | *Ahsan et al. (2023)* |
| | *Turnip mosaic potyvirus* and *Turnip yellow mosaic tymovirus* | Wild cabbage (*Brassica olerucea*) | Leaves | Reduces floral number and fruit set | *Maskell et al. (1999)* |
| | *Barley Yellow Dwarf Virus* | Winter wheat | Leaves | Reduces plant height and yield, and delays flowering | *Riedell et al. (1999)* |
| | | Maize | Leaves | Reduces plant and ear height, and accelerates flowering | *Körber (2013)* |
| | *Ageratum leaf curl Sichuan virus* | *Nicotiana benthamiana* | Leaves | Reduces plant height and delays flowering | *Li, Chen & Zhang (2022)* |
| Bacteria | *Erwinia amylovora* | Rosaceae plants | Anther | Causes floral withering | *Spinelli et al. (2005)* |
| | *Pseudomonas syringae* pv. *actinidiae* | Kiwifruit (*Actinidia chinensis*) | Anthers | Causes floral withering | *Donati et al. (2018)* |
| | *Pseudomonas syringae* and *Xanthomonas campestris* | Arabidopsis | Leaves | Accelerates flowering | *Korves & Bergelson (2003)* |

(Continued)

| Biological stress type | Stress source | Host plant(s) | Infection/ Feeding site | Impact/Damage | References |
|---|---|---|---|---|---|
| | *Burkholderia phytofirmans* | Arabidopsis | Roots | Accelerates flowering | *Poupin et al. (2013)*, *Esmaeel et al. (2018)* |
| | *Burkholderia seminalis* | Pepper and okra (*Abelmoschus esculentus* (L.) Moench) | Roots | Accelerates flowering and increases floral/fruit yield | *Hwang et al. (2021)* |
| | *Bacillus* sp. and *Mucilaginibacter* sp. | *Cannabis sativa* | Roots | Increases floral number | *Lyu, Backer & Smith (2022)* |
| Parasitic | *Struthanthus flexicaulis* | *Mimosa calodendron* | Stems | Reduces leaf area, fruit yield and seed weight | *Mourão et al. (2009)* |
| | *Rhinanthus serotinus* | *Linum usitatissimum* and *Brassica rapa* ssp. *oleifera* | Stems | Decreases floral and fruit number, shortens petals, and increases floral asymmetry | *Salonen & Lammi (2001)* |
| | *Cassytha filiformis* | *Suriana maritima*, *Scaevola plumieri*, and *Tournefortia gnaphalodes* | Stems | Reduces floral and fruit yield | *Parra-Tabla et al. (2024)* |
| | *Cuscuta partita* | *Zornia diphylla* | Stems | Reduces branch, leaf, floral, pollen and ovule number, and lowers pollen viability | *Cruz et al. (2017)* |
| | *Orobanche elatior* | *Centaurea scabiosa* | Roots | Reduces pollination efficiency and seed yield | *Ollerton et al. (2007)* |
| | *Orobanche* spp. | *Chrysanthemum morifolium* | Stems | Delays flowering or completely suppresses bloom | *Liu et al. (2021)* |
| Herbivores | *Megalurothrips sjostedti* | Cowpea (*Vigna unguiculata*) | Floral organs | Causes floral necrosis/abscission and prevents pod formation | *Alabi, Odebiyi & Tamò (2006)*, *Ngakou et al. (2008)* |
| | *Tanysphyrus lemnae* | *Sagittaria lancifolia* | Floral organs | Damages floral organs and reduces seed set per fruit | *Rodríguez-Morales, Aguirre-Jaimes & García-Franco (2024)* |
| | *Pieris brassicae* | Black mustard (*Brassica nigra* L.) | Floral organs | Damages floral organs | *Smallegange et al. (2007)* |
| | Chrysomelidae and Scarabaeidae | *Solanum rostratum* | Floral organs | Damages floral organs | *Gilmar-Moreira et al. (2022)* |
| | *Anthonomus signatus* | Wild strawberry (*Fragaria virginiana*) | Floral organs | Feeds on pollen | *Sõber, Moora & Teder (2010)* |
| | *Cionus nigritarsis* | *Verbascum nigrum* | Floral organs | Damages floral organs | *Penet, Collin & Ashman (2009)* |
| | *Pieris brassicae* and *Brevicoryne brassicae* | *Brassica nigra* | Leaves | Reduces floral number, accelerates flowering and decreases pollinator attractiveness | *Rusman et al. (2020)* |
| | *Bombus terrestris* | Tomato, black mustard (*Brassica nigra* L.) | Leaves | Accelerates flowering | *Pashalidou et al. (2020)* |

| Biological stress type | Stress source | Host plant(s) | Infection/ Feeding site | Impact/Damage | References |
|---|---|---|---|---|---|
| | *Aphis craccivora* Koch | Cowpea | Leaves | Delays flowering and reduces fruit yield | *Obopile (2006)*, *Obopile & Ositile (2010)* |
| | *Pieris rapae* | *Sinapis arvensis* | Leaves | Delays flowering | *Poveda et al. (2003)* |
| | *Agriotes* sp. | *Sinapis arvensis* | Roots | Increases nectar production and pollinator attraction | *Poveda et al. (2003)* |
| | *Acalymma vittatum* | Cucumber (*Cucumis sativus*) | Roots | Reduces leaf and fruit production, decreasing pollinator attraction | *Barber et al. (2015)* |
| | *Pieris c* | *Diplotaxis erucoides* | Leaves and Floral organs | Enhances the emission of floral volatiles that attract pollinators to improve pollination efficiency and lure natural enemies to limit floral damage | *Farré Armengol et al. (2015)* |
| | *Helicoverpa zea* | Cotton (*Gossypium hirsutum* L.) | Leaves and floral organs | Induces the release of volatile organic compounds that attract herbivore enemies | *Röse & Tumlinson (2004)* |
| | *Manduca sexta* | *Nicotiana attenuata* | Leaves | Induces the release of volatile organic compounds that attract both pollinators and herbivore enemies, thereby enhanceing pollination success and defense | *Zhou et al. (2017)* |
| | *Pieris brassicae* and *Spodoptera littoralis* | *Brassica rapa* | Leaves | Reduces the emission of floral volatiles that attract pollinators, resulting in decreased seed yield | *Schiestl et al. (2014)* |
| | *Manduca sexta* | *Solanum peruvianum* | Leaves | Causes significant changes in the volatile organic compound profile of floral tissues | *Kessler, Diezel & Baldwin (2010)* |
| | *Manduca sexta* and *Manduca quinquemaculata* | *Nicotiana attenuata* | Leaves | Decreases the release of floral volatiles that enhance attraction to both pollinators and herbivores | *Kessler, Diezel & Baldwin (2010)* |

**Table 1** (*continued*)

# SURVEY METHODOLOGY

To gather relevant literature for this manuscript, we conducted a comprehensive search using the Web of Science, PubMed databases, and reputable academic journals. Our literature retrieval process followed a two-step strategy. Initially, we performed the search using the combination of "biotic stress" AND "keywords related to plant reproductive development processes" (*e.g.*, flowering time, floral organ development, seed development, reproductive success, *etc.*). Subsequently, we conducted a more specific search by combining "particular biotic stress agents" (*e.g.*, fungi, viruses, bacteria, parasitic plants, herbivores) with the same set of keywords related to plant reproductive development. Articles retrieved were carefully evaluated based on their relevance to our topic. Articles that provided insights into how specific biotic stresses affect sexual reproduction in flowering plants and/or elucidated underlying mechanisms were downloaded for full-text reading. For selected references where the full texts were unavailable, we thoroughly examined their abstracts. Articles with unclear or ambiguous findings in their abstracts

were further excluded from our analysis. To ensure a balanced representation of the literature, we systematically considered studies regardless of publication date or journal impact factor, while preferentially citing recent or high-impact journal studies only when multiple studies reported consistent findings. We acknowledge that this selection process may introduce potential biases; however, our goal was to provide a comprehensive framework for understanding biotic stress impacts on plant reproduction. In our literature inclusion process, we did not exclude studies simply because they reported contradictory results. Instead, we systematically presented all relevant findings in the main text to provide readers with a comprehensive perspective. As this review aims to explore various biotic stresses affecting plant sexual reproduction and their underlying mechanisms through representative case studies, we carefully examined and synthesized both review articles and experimental studies, provided they contained relevant case examples that aligned with our thematic focus.

## IMPACT OF FUNGAL INFECTIONS ON PLANT REPRODUCTIVE DEVELOPMENT

A wide range of fungi, including *Fusarium*, *Alternaria*, *Fusicladium*, *Neoerysiphe*, *Mycosphaerella*, *Trichoderma*, and *Epicoccum*, has been identified worldwide (*Li et al., 2023*). The complex ecological interactions between these fungi and plants play a crucial role in shaping plant growth and development, with particularly significant impacts during the reproductive stage (*Bennett & Meek, 2020*; *Batzer et al., 2023*). During this critical phase, fungal infections can substantially alter flowering patterns, reduce fruit set, and compromise seed quality, ultimately affecting agricultural productivity (*Liu et al., 2018*; *Gonzalez-Gomez et al., 2021*; *Anand & Rajeshkumar, 2022*).

### Fungal infections influence flowering time

The flowering process, especially the timing of flowering, is integral to the successful reproduction of plants and the perpetuation of the species (*Chen et al., 2023*; *Li et al., 2024b*). Flowering time is intricately determined by an integrated regulatory network that emerges from the crosstalk between environmental cues and endogenous factors (*Li et al., 2022a*; *Jiang, 2023*). This intricate regulatory mechanism has rendered the study of flowering time a consistent area of research focus. Extensive research carried out on the plant model organism *Arabidopsis thaliana* (hereafter, Arabidopsis) and numerous other flowering plants has revealed the molecular mechanisms of five genetically defined pathways that regulate flowering, namely the photoperiod, autonomous, vernalization, gibberellin (GA), and age pathways (*Bao et al., 2020*; *Freytes, Canelo & Cerdán, 2021*; *Zhang et al., 2023*). These main pathways, governed by several key miRNAs, crosstalk with each other and ultimately converge on downstream floral integrator genes (involving *GIGANTEA* (*GI*), *CONSTANS* (*CO*), *FLOWERING LOCUS C* (*FLC*), *FLOWERING LOCUS T* (*FT*), *SUPPRESSOR OF OVEREXPRESSION OF CONSTANS 1* (*SOC1*), and *CRYPTOCHROME* (*CRY*)), which in turn transmit signals to downstream FM-identity genes (such as *APETALA1* (*AP1*), *FRUITFULL*, and *LEAFY* (*LFY*)), thereby orchestrating the process of flower formation (*Chen et al., 2018*; *Matar et al., 2021*; *Fan et al., 2022*).

Several excellent reviews have highlighted the significance of these pathways in regulating flowering (*Lee et al., 2023*; *Yang et al., 2024*; *Cai et al., 2024*). Additionally, increasing evidence shows that other phytohormones, such as abscisic acid (ABA), ethylene, cytokinin, salicylic acid (SA), auxin, and jasmonic acid (JA), also affect the flowering process (*Campos-Rivero et al., 2017*).

Fungal interactions with plants exhibit convergence in regulating flowering time through modulation of core flowering integrator genes, though pathogenic and symbiotic fungi may employ distinct strategies (Fig. 1A). The obligate biotroph *Peronospora parasitica* accelerates flowering in Arabidopsis by reducing the number of aerial branches while promoting early transition to reproductive phase (*Korves & Bergelson, 2003*). The pathogen *Fusarium oxysporum* directly targets the floral integrator *FLC*, a key repressor of flowering, while simultaneously inducing *FT* expression (*Lyons et al., 2015*). This pathogen additionally interferes with the photoperiod pathway as it modulates *GI* expression, which in turn activates *FT* (*Mizoguchi et al., 2005*). These coordinated changes in gene expression collectively alter plant physiology, thereby hastening the progression towards flowering (*Lyons et al., 2015*).

Endophytic fungi exhibit more complex interactions with flowering regulation (Figs. 1A and 2). *Piriformospora indica* establishes intercellular and intracellular colonization in Arabidopsis roots and systemically upregulates multiple flowering regulatory genes (*FT*, *LFY*, and *AP1*) and photoreceptor genes (*e.g.*, *CRY1* and *CRY2*) (*Kim et al., 2017*; *Pan et al., 2017*). This endophyte specifically activates GA biosynthetic genes (*e.g.*, *GA20ox2* and *GA3ox1*) and other GA-related genes (*e.g.*, *RGA1*, *AGL24*, *GA3*, and *MYB5*) while suppressing *FLC* expression, creating a permissive environment for flowering initiation (*Cheng et al., 2004*; *Kim et al., 2017*; *Pan et al., 2017*). The GA-dependence of this effect is evidenced by experiments where GA application promoted while GA inhibitors blocked *P. indica*-induced early flowering (*Pan et al., 2017*). *Pochonia chlamydosporia* employs a broader strategy, simultaneously activating multiple floral integrator genes *SOC1*, *FT*, *TWIN SISTER OF FT* (*TSF*), and *LFY* while suppressing *FLC* (*Zavala-Gonzalez et al., 2017*). This transcriptomic reprogramming may explain its strong effect on accelerating both flowering and fruiting in tomato (*Solanum lycopersicum*). The fungus likely achieves this through indole-3-acetic acid (IAA) production and phosphate solubilization, which indirectly affect flowering pathways (*Zavala-Gonzalez et al., 2015*).

These studies collectively reveal a recurring pattern that fungal infection (*e.g.*, *F. oxysporum*) or endophytic colonization (*e.g.*, *P. indica*, *P. chlamydosporia*) consistently downregulates the floral repressor *FLC* while activating positive regulators (such as *FT*, *SOC1*, and *LFY*) (Fig. 1A). The convergence of different fungal species on similar flowering regulators implies these genes represent key control points in the plant's development network that can be effectively targeted to alter flowering timing. Notably, while strongly influencing photoperiod and GA pathways fungal interactions leave age and autonomous pathways largely unaffected as shown by unchanged expression of their regulatory genes (*Pan et al., 2017*). However, the current understanding of these fungal-plant interactions is primarily based on targeted gene expression analyses, such as RT-qPCR, which focus on well-characterized flowering regulators (*Zavala-Gonzalez et al., 2017*; *Kim et al., 2017*;

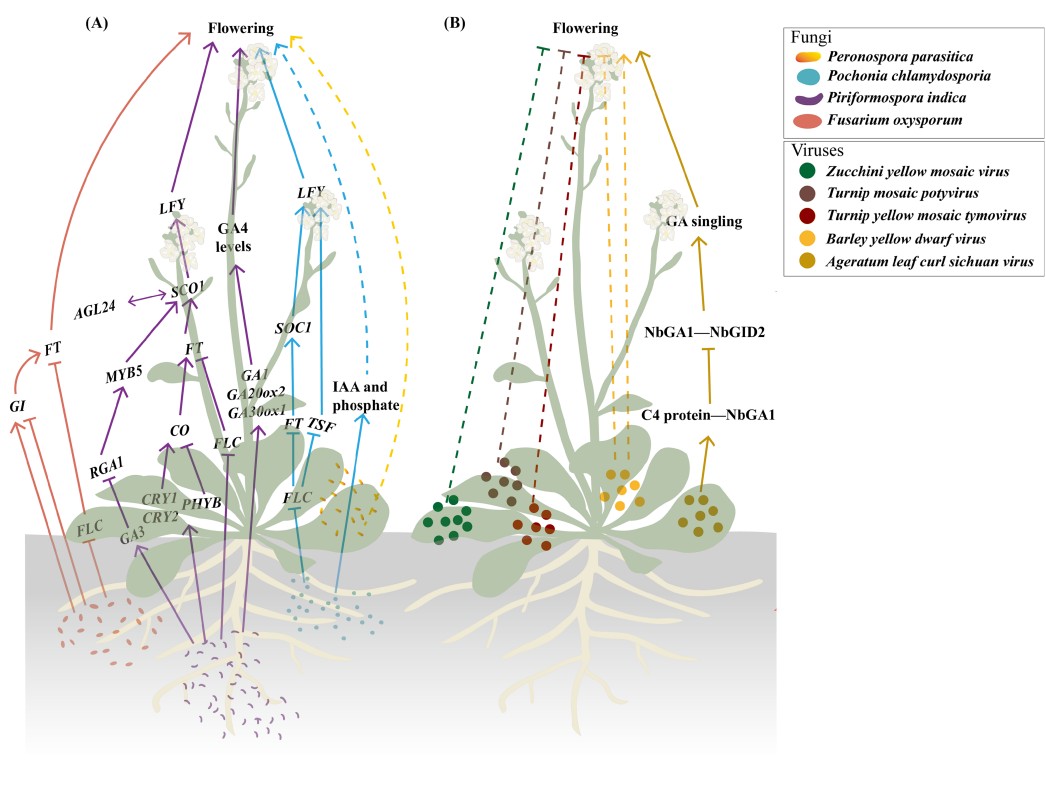

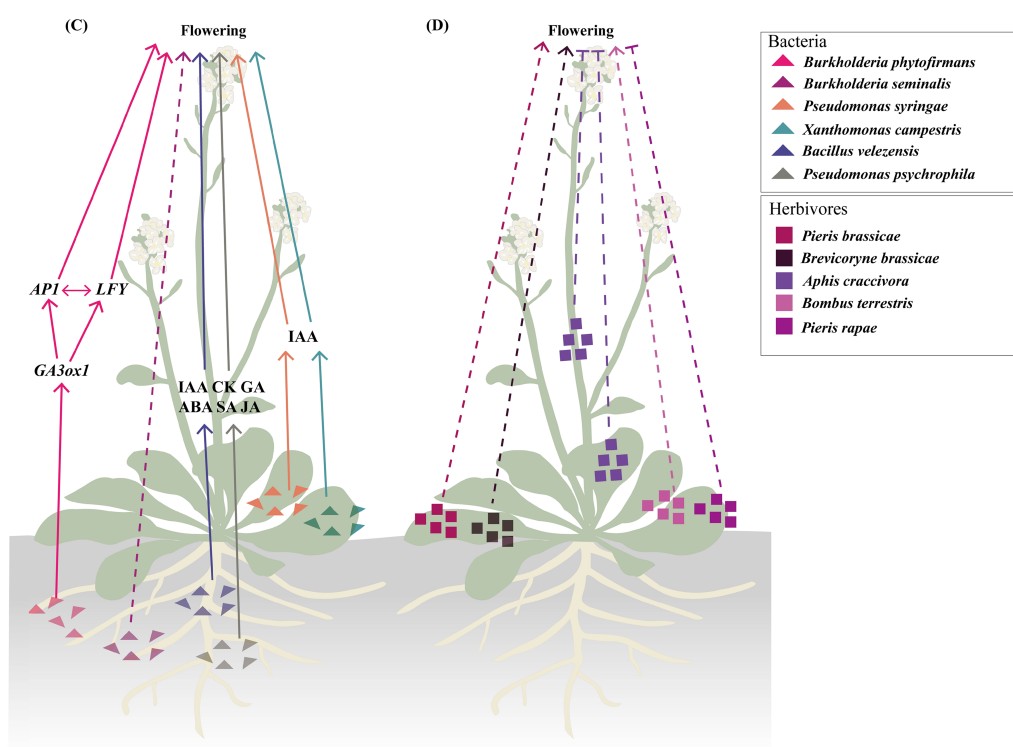

**Figure 1 Biotic stresses significantly affecting flowering time.** (A–C) Fungal (A), viral (B), and bacterial (C) infections involved in modulating flowering time. The fungal regulatory pathways on flowering illustrated in (A) integrate findings from studies (*Kumari et al., 2003*; *Korves & Bergelson, 2003*; *Martínez et al., 2004*; *Cheng et al., 2004*; *Mizoguchi et al., 2005*; *Das et al., 2012*; *Zavala-Gonzalez et al., 2015*, *2017*; *Lyons et al., 2015*; *Kim et al., 2017*; *Pan et al., 2017*; *Luo et al., 2019*). The regulatory

**Figure 1 (continued)**
pathway of Ageratum leaf curl Sichuan virus on flowering depicted in (B) is based on research by *Li et al. (2022b)*. The regulatory pathway of *Burkholderia phytofirmans* on flowering presented in (C) draws upon studies (*Poupin et al., 2013*; *Esmaeel et al., 2018*). (D) Herbivory behaviors that influence flowering time. Solid lines illustrate the already-determined regulatory mechanisms, with dashed lines representing the underlying mechanisms that remain unexplored. The figure was created offline using the Adobe Illustrator 2023 software (https://www.adobe.com/cn/creativecloud.html).

*Pan et al., 2017*). While these approaches robustly confirm the involvement of core flowering pathways, they may inadvertently overlook other potential mechanisms by which fungi modulate flowering time, such as uncharacterized genes or pathways, epigenetic modifications and post-translational regulation.

## Fungal infections impede floral organ development

While many fungi accelerate flowering to access nutrient-rich floral tissues (*e.g.*, *F. oxysporum*, *P. indica*), others delay flowering or disrupt reproductive development. *Ustilaginoidea virens*, a prominent pathogen in rice-cultivating regions worldwide and the causal agent of rice false smut (*Chen et al., 2022*), employs a contrasting strategy. This ascomycete fungus specifically infects the floral organs of rice (*Oryza sativa*), beginning with epiphytic growth on the lemma and palea, which are the outer and inner glumes that encase the floret, followed by an intercellular invasion of the stamen and pistil tissues (Fig. 2) (*Tang et al., 2013*; *Fan et al., 2020*). *Ustilaginoidea virens* primarily infects the stamen filaments, disrupting their development, preventing flowers from opening, and ultimately interrupting fertilization, which results in failed seed formation in the affected spikelets (*Fan et al., 2015*). Notably, unlike the root colonization by fungi such as *F. oxysporum*, *P. indica*, and *P. chlamydosporia*, which accelerate the floral transition by regulating key floral integrator genes (*Lyons et al., 2015*; *Zavala-Gonzalez et al., 2017*; *Kim et al., 2017*; *Pan et al., 2017*), *Ustilaginoidea virens* interferes with flower opening through disrupting stamen development. In the case of rice, this kind of disruption is part of the pathogen's mechanism to hijack the rice's reproductive structures to form false smut balls, which are a characteristic symptom of rice false smut disease (*Yu et al., 2023*). The divergent effects on flowering regulation highlight the complexity of plant-fungal interactions. While some fungi promote flowering through convergent modulation of key flowering genes (*Lyons et al., 2015*; *Zavala-Gonzalez et al., 2017*; *Kim et al., 2017*; *Pan et al., 2017*), *U. virens* suppresses it to maintain colonization sites. This dichotomy likely reflects distinct ecological strategies. Pathogens like *F. oxysporum* may accelerate flowering to exploit nutrient-rich tissues before disease progression and host death (*Lyons et al., 2015*), whereas *U. virens* prioritizes prolonged access to reproductive structures by inhibiting flower opening (*Fan et al., 2015*). Such differences underscore the importance of considering fungal lifestyle (pathogenic *vs*. endophytic) and infection site (roots *vs*. flowers) when interpreting their effects on plant development. To facilitate infection, *U. virens* secretes a cytoplasmic effector protein, UvCBP1, which interacts with the host protein OsRACK1A and competes with its binding to OsRBOHB. This interaction inhibits

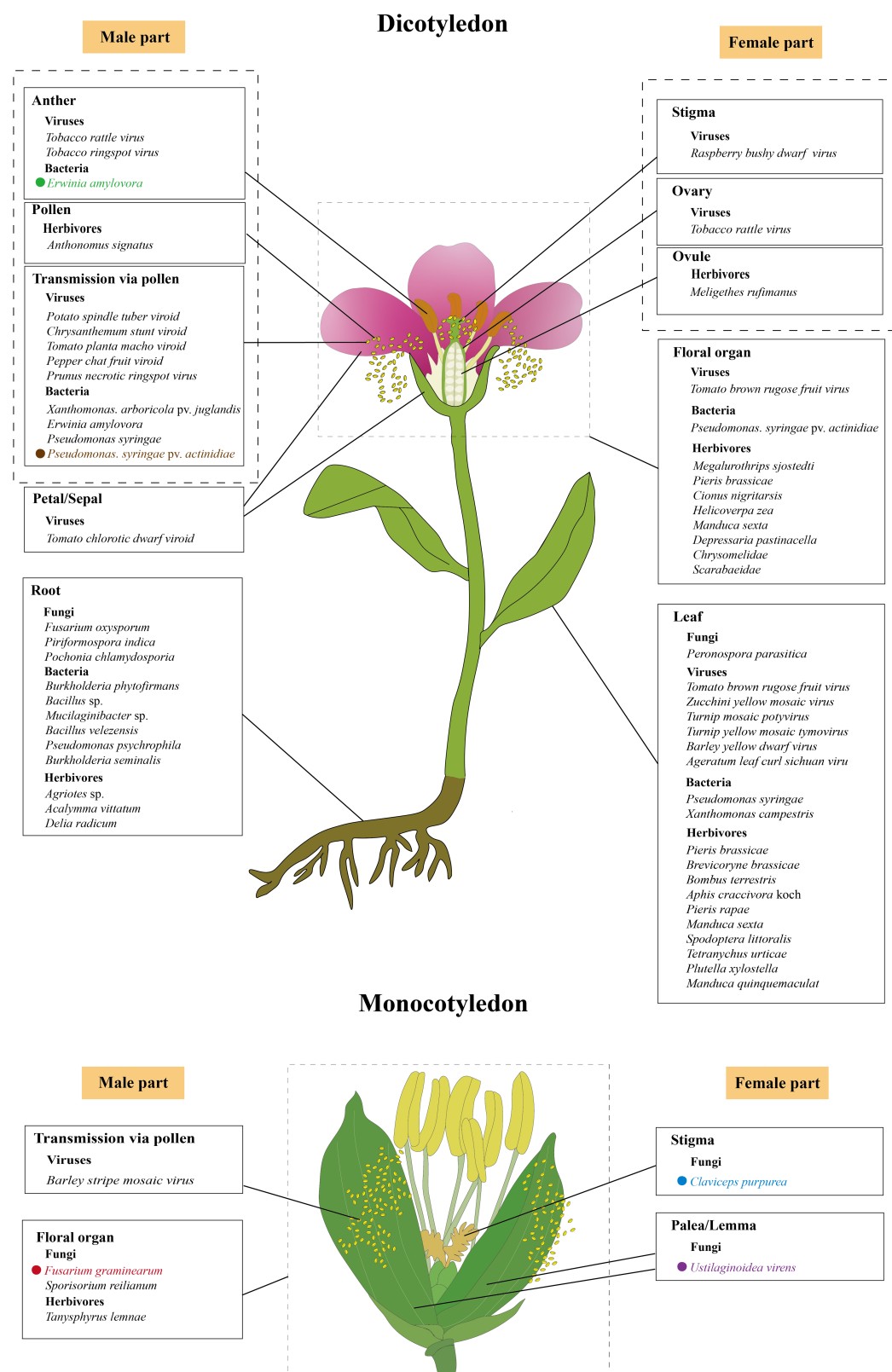

**Figure 2** **The initial infection/feeding sites of fungi, viruses, bacteria and herbivores that impact plant sexual reproduction.** The black solid lines demarcate the initial infection/feeding sites of various

**Figure 2 (continued)**
pathogens. Pathogens with colored fonts indicate that their infection pathways are established. *Erwinia amylovora* (green) colonizes the stamens, infects pollen, and subsequently spreads to the stigma *via* pollination before progressing along the stigma to the style and nectary (*Wilson, Sigee & Epton, 1989*; *Spinelli et al., 2005*). *Pseudomonas syringae* pv. *actinidiae* (brown) infects flowers through both stylar and anther pathways, colonizing either the stigma and style to reach the receptacle systemically or invading anthers to produce contaminated pollen for inter-plant transmission (*Donati et al., 2018*). *Fusarium graminearum* (red) establishes infection in floral bracts and ovaries before migrating downward through the rachis (*Pritsch et al., 2000*; *Wanjiru, Zhensheng & Buchenauer, 2002*). *Claviceps purpurea* (blue) colonizes the stigma before invading ovarian tissues to disrupt seed formation and development (*Miedaner & Geiger, 2015*; *Sun et al., 2020*; *Tente et al., 2021*). *Ustilaginoidea virens* (purple) penetrates the lemma and palea before compromising stamens and pistils, ultimately causing fertilization failure (*Tang et al., 2013*; *Fan et al., 2015*, *2020*). The figure was created offline using the Adobe Illustrator 2023 software (https://www.adobe.com/cn/creativecloud.html).

the production of reactive oxygen species (ROS), weakening the plant's immune response (*Li et al., 2022c*). In addition, the fungus secretes another virulence effector, SCRE4, into the nuclei of rice cells, where it downregulates the expression of the auxin response factor *OsARF17*, a key regulator in flower development, thereby indirectly disrupting the flowering process of rice (*Nagpal et al., 2005*; *Tabata et al., 2010*; *Qiu et al., 2022*). These molecular interventions contrast sharply with the gene activation strategies employed by flowering-promoting fungi, demonstrating how different fungal species have evolved distinct mechanisms to manipulate host development according to their ecological needs.

## Fungal infections affect seed development

Cereal crops such as rye (*Secale cereale*), barley (*Hordeum vulgare*), and wheat (*Triticum aestivum*) are especially vulnerable to infection by the fungus *Claviceps purpurea* during anthesis due to their open-flowering nature (*Mette et al., 2015*). The spores of *C. purpurea* infect female flowers by germinating on the unfertilized stigma and producing germ tubes that mimic the pollen tubes (Fig. 2), thereby circumventing host recognition mechanisms that are usually triggered by pollen-stigma interactions (*Sun et al., 2020*). Once inside the ovary, these mimicking germ tubes develop into a white, cottony mycelium. Over time, this mycelium condenses into a compact mass, culminating in the formation of a dark, hardened sclerotium, which replaces the developing seed and halts normal seed formation (*Miedaner & Geiger, 2015*; *Tente et al., 2021*).

## Fungal infections impact grain quality and crop yield

The mycotoxin-producing fungal pathogen *F. graminearum* is the primary cause of Fusarium head blight (FHB) in small grain cereals and cob rot of maize (*Zea mays*), leading to both substantial yield losses and reduced grain quality (*Johns et al., 2022*). Under natural conditions, FHB is typically initiated when airborne spores land on the flowering spikelets, with the open florets during the flowering phase serving as the initial entry point (*Boenisch & Schäfer, 2011*). In wheat, *F. graminearum* spores released from crop residues land on or inside the florets, where they germinate and initiate infection. The fungus penetrates floral tissues, spreading from the floral bracts and ovaries down through the rachis (Fig. 2) (*Pritsch et al., 2000*; *Wanjiru, Zhensheng & Buchenauer, 2002*). Most infections occur

during anthesis, partly because anthers contain stimulants for spore germination and pathogen growth (*Wegulo et al., 2015*). Another pathogenic fungus that invades the inflorescence tissues of wheat during flowering is the heterobasidiomycetous fungus *Ustilago tritici* (Persoon) Rostrup, which leads to the occurrence of loose smut accompanied by a significant reduction in the quality of grain seeds (*Thambugala et al., 2020*). *Ustilago tritici* normally survives as mycelium inside wheat seeds. Upon seed germination, the mycelium colonizes the crown node and subsequently invades the inflorescence. Once inside the florets, the spores germinate and develop into dikaryotic hyphae, which infect the ovaries and develop alongside the seed embryo (*Arif, 2019*). In maize, *Sporisorium reilianum* f. sp. zeae (Kühn) causes head smut, a systemic fungal disease that can reduce yields by up to 80% (*Jin et al., 2000*; *Zhou et al., 2022*). After infecting the plant, the fungus produces spores within the inflorescence (Fig. 2), leading to partial or complete replacement of the tassels and ears with large white galls filled with dark brown spores. The infection not only stunts growth and disrupts apical dominance but also triggers a variety of additional morphological abnormalities, contributing to significant losses in maize production (*Wang et al., 2024*).

## IMPACT OF VIRAL INFECTIONS ON PLANT REPRODUCTIVE DEVELOPMENT

### Viruses invade the reproductive organs

Plant viruses pose a significant threat to global agricultural productivity, ranking as the second most important group of plant pathogens after fungi (*Jaybhaye et al., 2024*). While many pathogenic viruses can infect the entire plant, most are unable to penetrate gametophytes, gametes, and/or progeny (embryos and seeds) (*Bradamante, Mittelsten-Scheid & Incarbone, 2021*). Plants have evolved meristematic and transgenerational antiviral defense systems that block viruses from being transmitted to the next generation (*Li et al., 2024a*). Nevertheless, under certain conditions, some viruses manage to bypass these defenses (*Bennett, 1969*).

The majority of plant viruses are transmitted through seeds, *via* both male and female gametes, with a higher frequency observed in pollen than in ovules (*Rajasekharan et al., 2024*). A recent comprehensive review highlighted three virus/viroid infection routes during plant sexual reproduction (*Bradamante, Mittelsten-Scheid & Incarbone, 2021*). One route is complete invasion, where a virus successfully infects all reproductive tissues, including gametes and embryos, enabling direct vertical transmission to the next generation (*Amari et al., 2007*). Another route is partial invasion, in which the viruses reach some reproductive tissues but do not infect the gametes or embryos. Although true vertical transmission does not occur, such infections can still be passed to seedlings post-germination *via* mechanical inoculation from the seed coat. For example, tomato chlorotic dwarf viroid has been observed in floral organs of tomato plants without reaching the embryos (Fig. 2) (*Matsushita, Usugi & Tsuda, 2011*). Moreover, transmission *via* pollen has been documented for a number of plant viruses and viroids, such as Barley stripe mosaic virus, Potato spindle tuber viroid, Chrysanthemum stunt viroid, Tomato planta macho viroid, and Pepper chat fruit viroid (Fig. 2) (*Bennett, 1969*; *Carroll, 1974*;

*Kryczyński, Paduch-Cichal & Skrzeczkowski, 1988*; *Mink, 1993*; *Card, Pearson & Clover, 2007*; *Woo, Clover & Pearson, 2012*; *Liu et al., 2014*; *Yanagisawa & Matsushita, 2017*; *Matsushita, Yanagisawa & Sano, 2018*). In these cases, a virus-infected pollen grain can fertilize a healthy plant, potentially infecting both the mother plant (horizontal transmission) and the developing embryo (vertical transmission) (*Matsushita, Usugi & Tsuda, 2011*; *Isogai et al., 2014*; *Matsushita, Yanagisawa & Sano, 2018*). Despite these observations, the underlying mechanisms governing these transmission pathways remain poorly understood.

## Viral infections negatively affect pollen performance

To date, at least 46 plant viruses have been reported to be pollen-transmitted (*Card, Pearson & Clover, 2007*; *Liu et al., 2014*). Some of these viruses have been demonstrated to impair pollen performance, resulting in reduced pollen quantity, lower viability, decreased germination rates, and inhibited pollen tube growth. For example, the Prunus necrotic ringspot virus (PNRSV) has been detected throughout the entire development of apricot (*Prunus armeniaca*) pollen after infection, from the pollen mother cells to the mature pollen and even in growing pollen tubes. Infected pollen shows a reduced germination rate and slower tube elongation, although it remains capable of fertilization (*Amari et al., 2007*). Notably, PNRSV can be transmitted *via* pollen to all reproductive organs, including embryos and even fruits (Fig. 2) (*Amari et al., 2009*). Similarly, the tomato brown rugose fruit virus (ToBRFV), a recently identified *Tobamovirus*, is abundantly present in various tissues, including the leaves, petals, stamens, styles, stigmas, pollen grains, and ovaries but not inside ovules (Fig. 2). Although ToBRFV-infected pollen appears normal in quantity and morphology, its ability to germinate is significantly impaired (*Avni et al., 2022*). In plants infected with Tobacco rattle virus (TRV), such as tobacco (*Nicotiana tabacum*) and pepper (*Capsicum annuum*), viral presence causes abnormalities in generative organs, leading to reduced flower and pollen production. Pollen grains from infected plants are often degenerated and have additional adverse effects on seed germination, seed quantity, and fruit formation, ultimately leading to reduced crop yields (*Otulak, Kozieł & Garbaczewska, 2016*). In *Torenia fournieri*, pollen tubes carrying Raspberry bushy dwarf virus (RBDV) are unable to progress beyond the style, halting fertilization (*Isogai et al., 2014*). Zucchini yellow mosaic virus (ZYMV), a member of the *Potyviridae* family, has caused severe losses in cucurbit crops throughout the world since the late 1970s (*Ahsan et al., 2023*). In wild squash (*Cucurbita pepo* subsp. *texana*), ZYMV infection results in reduced flower and fruit production per plant, decreased pollen production per flower, and lower overall fertility under competitive conditions (*Harth et al., 2016*). In soybean (*Glycine max*), anthers infected with tobacco ringspot virus produce fewer pollen grains with lower germination capacity and reduced pollen tube growth compared with those of healthy plants (Fig. 2) (*Yang & Hamilton, 1974*). Similarly, in highbush blueberries, infection by leaf mottle virus leads to reduced pollen quantity, smaller grain size, and diminished pollen viability (*Childress & Ramsdell, 1986*; *Madhavi, Rao & Subbarao, 2011*). The detrimental impacts of viral infections on pollen are also observed in a range of other crops, such as ring spot virus-infected papaya, Ilar virus-infected gherkin and okra, mosaic
virus-infected alfalfa, bottle gourd and chow-chow, yellow vein mosaic virus-infected okra, mosaic-infected watermelon and pumpkin, bud necrosis-infected watermelon, and yellow mosaic-infected bitter gourd flowers (*Pesic & Hiruki, 1988*; *Amari et al., 2007*; *Rajasekharan et al., 2024*). While these studies provide compelling evidence for virus-induced pollen dysfunction, the underlying molecular mechanisms remain largely unexplored. Current research has primarily relied on phenotypic observations and viral localization studies through techniques such as *in situ* hybridization, RT-qPCR, and immunogold labeling (*Amari et al., 2007*, *2009*; *Avni et al., 2022*), with limited investigation into the dynamic viral transmission processes and the specific molecular interactions between viral components and host pollen development.

## Viral infections influence the flowering process

The impact of viral infections on plant flowering has been well documented through numerous case studies (Fig. 1B). For instance, in courgette plants, inoculation with a mild strain of ZYMV delays flowering and consequently postpones fruit maturation, potentially reducing overall yield (*Spence et al., 1996*). Similarly, wild cabbage (*Brassica oleracea*) inoculated with either Turnip mosaic potyvirus (TuMV) or Turnip yellow mosaic tymovirus (TYMV) exhibits substantially reduced survival, stunted vegetative growth, and impaired reproductive development, accompanied by a diminished capacity to flower, fewer pods, and lower seed production (*Maskell et al., 1999*). In winter wheat, infection with barley yellow dwarf virus (BYDV) leads to reduced plant height, delayed anthesis, smaller seed size, and an overall decline in grain yield (*Riedell et al., 1999*). In contrast, BYDV-infected maize also shows growth inhibition characterized by reduced plant height and ear height, but interestingly, it flowers earlier than uninfected plants (*Körber, 2013*). This discrepancy may arise from the fundamental differences in the genetic regulation of flowering between winter wheat and maize. Winter wheat, a vernalization-requiring species, relies on prolonged cold exposure to initiate flowering (*Liu et al., 2024*). BYDV infection at the two-leaf stage (pre-vernalization) in wheat can disrupt resource allocation or signaling pathways critical for vernalization, delaying anthesis (*Riedell et al., 1999*). In contrast, maize, an annual plant without vernalization requirements, may prioritize stress-induced early flowering to ensure reproduction before viral damage escalates, a strategy observed in other stress responses (*Takeno, 2012*). These contrasting responses underscore the diverse effects that viral infection can have on the flowering process, although the underlying mechanisms remain poorly understood. Recent research has begun to clarify some of these mechanisms. In *Nicotiana benthamiana*, infection by Ageratum leaf curl Sichuan virus (ALCScV) delays flowering by interfering with GA signaling. Specifically, the viral C4 protein interacts with NbGAI, a negative regulator of GA signaling, disrupting its interaction with NbGID2. This interference inhibits the degradation of NbGAI, thereby suppressing GA signaling and resulting in delayed flowering and dwarfing (*Li et al., 2022b*).

These examples highlight how viral infections can significantly affect reproductive timing and success in plants. It is important to note, however, that plant viruses primarily pose a significant threat to crop production through early-stage infections or mixed infections (*Gaur et al., 2021*; *Navas-Castillo & Fiallo-Olive, 2021*).

# BACTERIA REGULATE PLANT REPRODUCTIVE DEVELOPMENT

## Pathogenic bacterial infections lead to flower withering and yield loss

Bacteria are among the most prevalent plant pathogens, with several species known for their high virulence and destructive impact on crops. Notable examples include *Ralstonia solanacearum*, *Pseudomonas syringae*, *Xanthomonas campestris*, *Xylella fastidiosa*, *Dickeya dadantii*, and some *Pectobacterium* species (*Mansfield et al., 2012*; *Gutiérrez-Pacheco et al., 2019*). These pathogens collectively contribute to substantial reductions in fruit quality and yield, affecting approximately 10% of global crops during both pre- and post-harvest stages (*Din, 2011*). Most bacteria, such as *X. oryzae* pv. *oryzae* (*Xoo*), *X. axonopodis* pv. *citri*, *X. fastidiosa*, and *Candidatus Liberibacter asiaticus*, reduce crop yields by damaging roots and/or leaves, thereby impairing water and nutrient uptake (*Graham et al., 2004*; *Vojnov et al., 2010*; *Wang et al., 2017*; *Ference et al., 2018*; *Oliva et al., 2019*; *Ahmed et al., 2020*). Pollen-mediated transmission of bacterial pathogens is a relatively rare phenomenon, documented only for a few species such as *X. arboricola* pv. *juglandis*, *Erwinia amylovora*, and *P. syringae* (Fig. 2) (*Ercolani, 1962*; *Wilson, Sigee & Epton, 1989*; *Mansvelt & Hattingh, 2011*). Among them, *X. arboricola* pv. *juglandis*, which specifically infects walnuts (*Juglans regia* L.), is disseminated by wind and rain and can colonize pollen, enabling it to spread to healthy plants *via* pollination (*Kałużna et al., 2021*). In Rosaceae plants, *E. amylovora* inoculated onto the stamens of freshly opened flowers can penetrate the anther locules through the dehiscence zone and subsequently infect the pollen grains (*Wilson, Sigee & Epton, 1989*). Following infection, the pathogen then colonizes the stigmas *via* pollination, subsequently migrating towards the nectaries along a stylar groove lined with papillae (*Spinelli et al., 2005*). Recent studies in kiwifruit (*Actinidia chinensis*) have detailed the floral infection and pollen-mediated spread of *P. syringae* pv. *actinidiae* (*Psa*), the causative agent of kiwifruit bacterial canker. This pathogen causes typical flower symptoms, including browning of petals and sepals in the early stages of infection, followed by flower withering before blooming or shortly after fruit set. *Psa* invades flowers through two primary pathways: the stylar pathway and the anther pathway (Fig. 2). In the stylar pathway, it colonizes the stigma, travels along the stylar furrow, and enters the receptacle *via* the style or nectar grooves. From there, the bacteria migrate into the pedicel and become systemic. In the anther pathway, *Psa* colonizes anthers both epiphytically and endophytically, producing infected pollen that can transmit the pathogen to healthy plants through fertilization (*Donati et al., 2018*). Likewise, other species, such as *P. viridiflava* and *P. syringae* pv. *syringae*, are also known to cause floral blight and similar symptoms (*Young et al., 1988*; *Spinelli et al., 2005*; *Donati et al., 2020*).

## Plant growth-promoting bacteria impair sexual reproduction by modulating the balance of phytohormones

Phytohormones are well-established as key regulators of sexual reproduction in plants, playing vital roles in processes such as floral primordia differentiation, flowering induction, stamen and pollen development, seed setting, and fruit development. Extensive reviews have discussed the roles of various phytohormones, such as auxin, cytokinin, GA,

ABA, SA, and JA, in these processes (*Barazesh & McSteen, 2008*; *Pagnussat et al., 2009*; *Kaur et al., 2021*; *Castro-Camba et al., 2022*; *Huang et al., 2023*).

Over the past decades, compelling evidence has emerged indicating that plant growth-promoting bacteria (PGPB) can influence the sexual reproductive development of flowering plants by modulating the biosynthesis or degradation of phytohormones (*Nascimento, Glick & Rossi, 2021*; *Asif et al., 2022*). For example, in Arabidopsis, infection with the biotrophic bacterium *P. syringae* or *X. campestris* (Fig. 2) resulted in a dose-dependent reduction in both flowering time and the number of aerial branches on the primary inflorescence (*Korves & Bergelson, 2003*). These changes were linked to a notable increase in IAA accumulation (Fig. 1C) (*O'Donnell et al., 2003*). Inoculation with two PGPR strains, *Bacillus* sp. and *Mucilaginibacter* sp. (Fig. 2), was found to promote flower production and enhance axillary bud outgrowth (*Lyu, Backer & Smith, 2022*). More recently, three PGPR strains, namely *B. velezensis* RI3 and SC6 and *P. psychrophila* P10 (Fig. 2), were shown to significantly boost flower number, flowering rate, seed quality, and yield in peanut (*Arachis hypogaea* L.). These enhancement effects were attributed to elevated concentrations of IAA, cytokinin, GA, ABA, SA, and JA (Fig. 1C) (*Bigatton et al., 2024*). Additionally, rhizosphere microbial communities that enhance and prolong nitrogen bioavailability have been found to convert tryptophan to IAA, a process that can delay flowering (*Lu et al., 2018*; *Lyu, Backer & Smith, 2022*). It is important to note that while certain phytohormones may predominantly regulate specific reproductive processes, the coordination and balance among multiple hormones are essential for their coordinated functions to ensure proper reproductive development (*Mukherjee et al., 2022*).

## Endophytic bacteria improve flowering

Endophytic bacteria, which reside within plant tissues and can be isolated from surface-sterilized plant tissues without causing disease, play a significant role in enhancing nutrient uptake and promoting plant growth, particularly in accelerating flowering (*Afzal et al., 2019*). In Arabidopsis, inoculation with *Burkholderia phytofirmans* PsJN, a well-known plant endophytic bacterium and a plant growth-promoting rhizobacterium (PGPR) (Fig. 2), was shown to shorten the time to flowering and induce early signs of senescence compared with the non-inoculated controls. Transcriptome analysis further revealed that PsJN inoculation triggers the upregulation of *GA3ox1* and early activation of the meristem identity genes *LFY* and *AP1* (*Poupin et al., 2013*; *Esmaeel et al., 2018*). Similarly, switchgrass (*Panicum virgatum* L.) inoculated with PsJN also exhibits earlier leaf senescence and accelerated flowering (*Wang, Seiler & Mei, 2015*). Another example includes the plant endophytic bacterium *B. seminalis* strain 869T2 (Fig. 2), which enhances flower and fruit production in pepper and promoted both earlier flowering and increased fruit weight in okra (*Abelmoschus esculentus* (L.) Moench) (*Hwang et al., 2021*). Although these findings underscore the potential of endophytic bacteria to influence the timing of flowering (Fig. 1C), the underlying mechanisms remain largely unexplored.

# IMPACTS OF PARASITIC PLANTS ON SEXUAL REPRODUCTION PROCESS

Many parasitic plants constitute important agricultural weeds, and these weeds, such as dodders (*Cuscuta* spp.), witchweed (*Striga* spp.), and broomrapes (*Orobanche* spp.), pose a serious threat to crop productivity worldwide (*Zagorchev et al., 2021*). Recent estimates indicate that there are approximately 4,750 known parasitic plant species within the angiosperms, spanning 292 genera (*Nickrent, 2020*). These species employ a wide range of parasitic strategies, ranging from hemiparasitism with retained photosynthetic capability to holoparasitism characterized by an absolute nutritional reliance on the host organisms for survival. Through specialized adaptations, parasitic plants form intimate connections with their hosts to extract water, nutrients, and photosynthates, often weakening host vitality and reproductive performance (*Cruz et al., 2017*).

## Parasitic plants impair host flower and fruit production

Parasitic plants employ a specialized structure called a haustorium to attach to and penetrate host tissues, establishing vascular connections. This allows them to siphon off water, nutrients, and even secondary metabolites and proteins from the hosts. Such resource diversion can severely impair the host's vegetative growth and reproductive output, resulting in fewer, smaller, and less viable flowers and seeds (*Teixeira-Costa & Davis, 2021*). Both hemiparasitic and holoparasitic plants can significantly impede the development of reproductive organs in host plants (*Hibberd et al., 1996*; *Fernandes et al., 1998*; *Puustinen & Salonen, 1999*; *Irving & Cameron, 2009*; *Mourão et al., 2009*; *Bahia et al., 2015*). For example, parasitism of *Mimosa calodendron* by the hemiparasitic plant *Struthanthus flexicaulis* was found to result in a substantial reduction in host leaf area, lower fruit production, and decreased seed weight (*Mourão et al., 2009*). *Phoradendron californicum*, a desert mistletoe, can reduce fruit yield in desert trees by limiting both the nutrition and photosynthetic area of the host plant (*Yule, 2018*). Similarly, *Linum usitatissimum* and *B. rapa* ssp. *oleifera* infected with the hemiparasitic plant *Rhinanthus serotinus* also exhibit reduced flower and fruit numbers, shortened petals, and increased floral asymmetry (*Salonen & Lammi, 2001*). These changes can lead to decreased pollination efficiency and ultimately affect the reproductive success of the host plants. Interestingly, while *Cassytha filiformis* negatively impacts the flower and fruit production in three host species (*Suriana maritima*, *Scaevola plumieri*, and *Tournefortia gnaphalodes*), it enhances reproductive success in *S. maritima*—an exception that suggests host-specific outcomes (*Parra-Tabla et al., 2024*). In another case, parasitism of *Zornia diphylla* by the holoparasitic plant *Cuscuta partita* can also significantly undermine both the vegetative and reproductive characteristics of the host, including fewer branches, leaves, and flowers; reduced quantities of pollen and ovules; and lower pollen viability (*Cruz et al., 2017*). In addition, parasitic plants may indirectly influence seed dispersal. For instance, fruits of the mistletoe *P. juniperinum* can enhance bird-mediated seed dispersal of its host *Juniperus monosperma*, highlighting the complex ecological interactions involving parasitic plants, hosts, and animal vectors (*Ommeren & Whitham, 2002*).

## Parasitic plants and hosts interact reciprocally in flowering

In ecological systems, parasitic plants and their hosts exist in an involuntary state of coexistence. When the plants attract the same pollinators during flowering, their pollination niches may overlap. This overlap includes factors such as flowering time, pollinator species, and the timing of pollen and stigma availability (*Heithaus, 1974*; *Hansen, Armbruster & Antonsen, 2000*; *Ollerton et al., 2003*, *2007*). For example, the specialist parasitic plant *Orobanche elatior* (Orobanchaceae) and its host *Centaurea scabiosa* (Asteraceae) flower at roughly the same time, resulting in a shared pollination niche. This overlap can lead to competition for pollinators such as *Bombus pascuorum*, ultimately reducing pollination efficiency and seed production (*Ollerton et al., 2007*). In the case of chrysanthemums (*Chrysanthemum morifolium*), infection by dodder leads to stunted growth, yellowing and drying of leaves, and, in severe cases, plant death. Prolonged parasitism often results in delayed flowering or complete floral suppression, severely diminishing the ornamental value of chrysanthemums (*Liu et al., 2021*). The immune response of chrysanthemum to dodder was speculated to possibly involve complex signaling pathways related to ROS, calcium, ethylene, and SA signaling (*Liu et al., 2021*), although the precise interaction mechanisms are not yet fully understood.

Interestingly, host plants can also influence the flowering behavior of parasitic plants. Notably, certain dodder species, such as *C. australis*, can synchronize their flowering with that of their hosts. In crops like soybean and tobacco, this synchronization is driven by the ability of *C. australis* to "eavesdrop" on the host-derived FT proteins. These proteins migrate into the dodder's stem, where they interact with the FD transcription factor to activate flowering in the dodder (*Shen et al., 2020*). However, this mechanism is not universal among dodder species. For instance, *C. campestris* does not exhibit the same floral response after parasitizing its host. A recent study in tobacco further demonstrated that the host's flowering status, specifically in $Ntft4^-Ntft5^-$-double-knockout mutants and $NtFT5$-overexpressing plants ($35S:NtFT5_{L4}$//SR1$\Delta NtFT5$), does not significantly change the flowering time of *C. campestris*, suggesting that *C. campestris* does not rely on the host's FT signaling pathway to initiate flowering (*Mäckelmann et al., 2024*). It is hypothesized that host-mediated effects contribute to reproductive phenological asynchrony in parasitic plants, which can influence pollination success, seed dispersal, offspring quality, and the animals that depend on these plants (*Li, Chen & Zhang, 2022*). For instance, a study on the host-mediated effects on the generalist mistletoe *Dendrophthoe pentandra* (Loranthaceae) found that different host species can alter the duration of mistletoe flowering and fruiting, leading to phenological mismatches (*Li, Chen & Zhang, 2022*). Similarly, a study on the desert mistletoe (*Phoradendron californicum*) revealed that plants parasitizing mesquite (*Prosopis velutina*) produce more pollinator rewards per flower and receive more pollen grains per flower compared with those parasitizing catclaw acacia (*Senegalia greggii*), although fruit production remains similar across hosts (*Yule & Bronstein, 2018*). These host-driven variations in the reproductive phenology of parasitic plants can either promote or diminish reproductive isolation among populations, highlighting the complex ecological interactions between parasitic plants and their hosts.

# THE MULTIFACETED INFLUENCE OF HERBIVORES ON PLANT REPRODUCTION

Herbivory is intricately linked to plant reproduction. Herbivores can directly reduce the reproductive success of a plant by damaging flowers (florivory) or indirectly influence it through damage to the vegetative parts such as leaves, stems, or roots (folivory and other forms of vegetative herbivory) (*Poveda et al., 2003*; *McCall & Irwin, 2006*). These forms of damage can alter key floral traits, affecting the plant's overall floral display. As a result, pollinator visitation patterns may shift, ultimately affecting the efficiency of pollination and the success of the plant's reproduction.

## Herbivores directly damage floral tissue through florivory

Flowers, being one of the plant's primary nutritional reservoirs, often have higher nutritional value than leaves, making them particularly attractive to herbivorous animals (*Haan, Bowers & Bakker, 2021*). As a result, florivory can directly impact seed production by damaging key reproductive structures (*Alabi, Odebiyi & Tamò, 2006*). For instance, thrips (*Megalurothrips sjostedti*) are a major pest of cowpea (*Vigna unguiculata* (L.) Walp.), attacking the crop from the pre-flowering to flowering stages (Fig. 2). Their feeding causes necrosis and/or abscission of buds and flowers, ultimately preventing pod formation and causing substantial yield losses (*Alabi, Odebiyi & Tamò, 2006*; *Ngakou et al., 2008*). The larvae of cabbage white butterflies (*Pieris brassicae*) also exhibit specific feeding behaviors. Starting in the late second instar stage, they feed exclusively on the flower buds and flowers of black mustard (*B. nigra* L.) (Fig. 2), likely due to the higher concentration of glucosinolates in these tissues (*Smallegange et al., 2007*). Florivory is also common in *Sagittaria lancifolia*, where insects like the weevil *Tanysphyrus lemnae* damage flowers (Fig. 2), reducing their attractiveness and lowering seed production per fruit (*Rodríguez-Morales, Aguirre-Jaimes & García-Franco, 2024*). In other species, florivores directly consume ovules, as seen in *Isomeris arborea*, where this leads to decreased seed output (*Krupnick & Weis, 1999*). Other examples include beetles from the *Chrysomelidae* and *Scarabaeidae* families, which consume floral parts such as the corolla, anthers, and stamens in *S. rostratum* (Fig. 2) (*Gilmar-Moreira et al., 2022*). The larvae of *Anthonomus signatus* feed on the pollen of wild strawberry (*Fragaria virginiana*), while *Cionus nigritarsis* larvae consume the floral and reproductive tissues of *Verbascum nigrum* (Fig. 2) (*Penet, Collin & Ashman, 2009*; *Sõber, Moora & Teder, 2010*). These examples illustrate the widespread and varied nature of florivory. The impact on plant reproduction depends on both the specific floral organs consumed and the plant's reproductive system (*Cárdenas-Ramos & Mandujano, 2019*). In hermaphroditic and self-compatible plants, the removal of petals by florivorous insects may increase the chances of self-pollination (*Penet, Collin & Ashman, 2009*). However, in insect-pollinated species such as dioecious or monoecious species, florivory can significantly decrease reproductive success (*Hillier, Evans & Evans, 2018*; *Boaventura et al., 2022*; *Jabbour et al., 2022*).

## Herbivores impact reproductive processes through vegetative herbivory

Herbivores can affect plant reproductive processes not only by consuming floral parts but also through vegetative herbivory, which targets non-reproductive structures such as leaves and roots (Fig. 2). Damage to these vegetative organs disrupts energy accumulation and the synthesis of vital compounds (Barber et al., 2015; de Vries et al., 2018), ultimately limiting the plant's ability to support reproductive development (Strauss, Conner & Rush, 1996; Poveda et al., 2005). These indirect effects can influence key aspects of reproduction, including reductions in flower number and size, changes in floral morphology, shifts in flowering phenology, and a decrease in the production of floral rewards like nectar and pollen, ultimately impacting pollinator behavior (Strauss, Conner & Rush, 1996; Lehtilä & Strauss, 1999; Mothershead & Marquis, 2000; Poveda et al., 2003; Hanley & Fegan, 2007; Samocha & Sternberg, 2010; Kessler, Halitschke & Poveda, 2011; Schiestl et al., 2014; Bruinsma et al., 2014). For example, in *B. nigra*, leaf feeding by larvae of *P. brassicae* and the aphid *Brevicoryne brassicae* during the vegetative stage reduces the number of flowers and promotes earlier flowering (Fig. 1D), thereby lowering pollinator attraction (Rusman et al., 2020). Similarly, in tomato and black mustard plants that have not yet flowered, leaf damage by bumblebees (*Bombus terrestris*) significantly accelerates flowering when pollen is limited (Fig. 1D) (Pashalidou et al., 2020). In contrast, in cowpea, herbivory by the cowpea aphid (*Aphis craccivora* Koch) can significantly impede plant growth, delay the onset of flowering (Fig. 1D), and, in severe cases, reduce yield by over 50% (Obopile, 2006; Obopile & Ositile, 2010). In *Sinapis arvensis*, leaf herbivory by cabbageworms (*P. rapae*) reduces photosynthetic capacity, decreasing resource allocation to inflorescences and delaying flowering (Fig. 1D). Interestingly, root herbivory by wireworms (*Agriotes* sp.) in the same species increases nectar production, thereby attracting more pollinators. However, when both roots and leaves are simultaneously subjected to herbivory, severe losses in photosynthetic and root function lead to a shortened flowering period (Poveda et al., 2003). Some herbivory responses may even increase reproductive output. For instance, *Raphanus* plants attacked by *P. rapae* tend to produce more flowers, which can enhance their male fitness (Strauss, Conner & Lehtilä, 2001). Furthermore, in cucumber (*Cucumis sativus*), intense root herbivory by *Acalymma vittatum* reduces both leaf and fruit production and leads to decreased pollinator visitation (Barber et al., 2015).

## Herbivore-induced plant volatiles increase the reproductive success

When florivorous animals damage plants, they often trigger the release of herbivore-induced plant volatiles (HIPVs), which serve as airborne chemical signals that can influence the behavior of neighboring plants, herbivores and the natural enemies of herbivores (Whitman, 1990; Smallegange et al., 2007; Dicke, 2009; Hopkins, Dam & Loon, 2009; Zangerl & Berenbaum, 2009). For example, when larvae of the cabbage white butterfly (*P. brassicae*) feed on the flowers and leaves of wall rocket (*Diplotaxis erucoides*) (Fig. 2), the plant markedly increases its release of three volatile organic compounds (VOCs): methanol, 3-butenenitrile, and ethyl acetate. These VOCs serve a dual purpose—attracting pollinators to enhance pollination efficiency, while simultaneously

luring natural enemies of the herbivores to limit floral damage (*Farré Armengol et al., 2015*). Similarly, when *Helicoverpa zea* larvae feed on the flower buds of cotton (*Gossypium hirsutum* L.) (Fig. 2), they induce the release of a variety of terpenoid compounds from both the damaged buds and from nearby undamaged leaves. These VOCs attract the natural enemies of *H. zea* larvae, protecting the flower buds from further damage (*Röse & Tumlinson, 2004*). In *N. attenuata*, when *Manduca sexta* feeds on the leaves during the flowering stage (Fig. 2), the plant releases the volatile compound (*E*)-α-bergamotene from both its leaves and flowers. Leaf release attracts predatory insects that prey on *M. sexta* larvae, whereas flower release attracts pollinators, thereby enhancing both defense and pollination success (*Zhou et al., 2017*). In addition, herbivore frass can also influence floral scent. In wild parsnip (*Pastinaca sativa*), the frass of the parsnip webworm (*Depressaria pastinacella*) contains n-octanol, a metabolite of octyl ester metabolism. This compound alters the composition of the plant's floral volatiles, affecting how attractive the flowers are to pollinators (*Zangerl & Berenbaum, 2009*). These examples highlight how HIPVs, while initially triggered by damage, can have adaptive benefits—both by defending reproductive structures and enhancing pollination.

Just as plants release HIPVs in response to florivory, they also release these HIPVs following vegetative herbivory. In *B. rapa*, for instance, leaf feeding by *P. brassicae* and *Spodoptera littoralis* (Fig. 2) reduces the content of floral VOCs, making the flowers less attractive to pollinators and ultimately diminishing seed production (*Schiestl et al., 2014*). When herbivores feed on the leaves of brassicaceous plants and tomatoes, the plants release large amounts of the homoterpene 4,8-dimethyl-1,3,7-nonatriene during the flowering stage (*Kant et al., 2004*; *Soler et al., 2007*; *Abel et al., 2009*). This important HIPV plays a dual role: it not only attracts the natural enemies of herbivores, protecting the plants against further damage, but can also potentially influences the behavior of pollinators (*Kappers et al., 2005*; *Mumm & Dicke, 2010*). In *S. peruvianum*, leaf feeding by *M. sexta* larvae causes significant changes in the VOC profile of floral tissues, notably altering the release of (*E*)-α-bergamotene and benzylacetone (BA). Similarly, when *M. quinquemaculata* and *M. sexta* feed on *N. attenuata* leaves, BA release from the flowers decreases, which paradoxically increases flower attractiveness to nocturnal hawkmoths (*M. quinquemaculata* and *M. sexta*), serving as both pollinators and herbivores (*Kessler, Diezel & Baldwin, 2010*). These examples illustrate how HIPVs triggered by vegetative herbivory can protect plant reproductive structures by attracting predators of herbivores while also modulating pollinator interactions. This multi-layered defense ultimately enhances the plant's reproductive success (*Kessler & Halitschke, 2009*).

## CONCLUSIONS AND PROSPECTS

Plant reproductive development is precisely regulated by a complex network of environmental cues and internal factors. In recent years, there has been considerable interest in understanding how biotic stresses such as fungi, bacteria, viruses, parasitic plants, and herbivores affect this critical process. Here, our review reveals distinct yet overlapping mechanisms through which these stressors influence plant reproduction. Fungal pathogens demonstrate remarkable tissue specificity in their attacks.

Root-colonizing species like *P. indica* and *P. chlamydosporia* systemically accelerate flowering through phytohormonal manipulation and direct regulation of flowering genes (*Cheng et al., 2004*; *Zavala-Gonzalez et al., 2015*, *2017*; *Kim et al., 2017*; *Pan et al., 2017*). In contrast, floral-infecting fungi such as *U. virens* and *C. purpurea* employ more localized strategies, directly disrupting gametophyte development and seed formation through effector proteins and physical replacement of reproductive structures (*Sun et al., 2020*; *Li et al., 2022c*). Viruses exhibit unique transmission strategies that differentiate them from other biotic stressors. While most are excluded from meristematic tissues, viruses like PNRSV and ToBRFV exploit pollen as transmission vectors, often impairing pollen viability and tube growth (*Amari et al., 2007*, *2009*; *Avni et al., 2022*). Their systemic nature allows them to alter flowering time through disruption of phytohormone signaling pathways. Bacterial pathogens, represented by species such as *P. syringae* and *X. campestris*, typically cause broad physiological disturbances rather than targeted reproductive attacks (*Korves & Bergelson, 2003*). Their impact on reproduction is often secondary to systemic effects on plant health, though some species like *E. amylovora* have evolved specialized mechanisms for floral infection and pollen-mediated transmission (*Wilson, Sigee & Epton, 1989*). Bacterial effectors frequently manipulate auxin signaling, creating imbalances that affect flowering time and floral development. Parasitic plants occupy a unique ecological niche, with species like *Cuscuta* and *Striga* employing haustoria to directly tap into host vascular systems (*Zagorchev et al., 2021*). This intimate association allows for sophisticated manipulation of host physiology, including synchronization of flowering times through interception of FT signals. The resource drain imposed by parasitic plants often leads to dramatic reductions in flower number and seed set. Herbivores exert their influence through both direct consumption and induced physiological changes. Florivores such as thrips and lepidopteran larvae cause immediate damage to reproductive structures, while folivores induce systemic changes in floral traits and volatile profiles that alter pollinator behavior (*Alabi, Odebiyi & Tamò, 2006*; *Ngakou et al., 2008*; *Schiestl et al., 2014*; *Barber et al., 2015*; *de Vries et al., 2018*). The production of HIPVs represents a sophisticated defense mechanism that can simultaneously attract natural enemies of herbivores while modifying pollinator attraction (*Schiestl et al., 2014*; *Zhou et al., 2017*). Despite these differences, common themes emerge across stress categories. All biotic stressors ultimately influence reproductive success through modulation of phytohormone pathways, particularly GA, auxin, and JA. Additionally, many have evolved mechanisms to either accelerate or delay flowering time to align with their life cycles. The convergence on these core regulatory networks suggests they represent vulnerable nodes in plant reproductive development that are frequently targeted by diverse biotic stressors.

While current research has made significant progress in elucidating the effects of these biotic stresses on sexual reproduction in flowering plants, the underlying molecular mechanisms remain largely unclear. These knowledge gaps present critical research directions for future investigations, particularly in elucidating how biotic stressors influence reproductive development through transcriptional and epigenetic regulation. Emerging evidence highlights the importance of non-coding RNAs (*e.g.*, lncRNAs and

miRNAs) in mediating plant responses to biotic stresses (*Yajnik, Singh & Singh, 2024*). The Arabidopsis lncRNA *SABC1*, for instance, fine-tunes SA biosynthetic pathway during *P. syringae* infection, balancing defense responses with growth requirements (*Liu et al., 2022*). Another compelling case involves Osa-miR535 in rice, which regulates blast disease resistance by targeting the *OsSPL4-GH3.2* regulatory module, demonstrating how miRNAs can orchestrate immunity through post-transcriptional control (*Zhang et al., 2022*). In wheat, an extensive network comprising 590 miRNA-lncRNA interactions coordinates defense against *Puccinia graminis* f. sp. *tritici* (*Pgt*) infection by modulating resistance-related genes (*Jyothsna, Nair & Alagu, 2025*). Nonetheless, the specific roles of these regulatory elements in reproductive development, such as the onset of flowering, formation of floral organs, and development of gametophytes under biotic stresses, remain poorly understood and warrant systematic investigation.

These mechanistic insights inform practical strategies for crop improvement. For instance, Fungal and bacterial pathogens often interfere with reproductive success by manipulating key flowering genes such as *FLC*, *FT*, and *SOC1*, suggesting CRISPR-based editing of these loci could generate stress-resilient varieties. The success of *Pijx* gene-introgressed blast-resistant rice, which increased yields up to 79% (*Xiao et al., 2023*), demonstrates the potential of such strategies. For viral pathogens, RNA interference (RNAi) strategies targeting viral genomes or host susceptibility factors like NbGAI in GA signaling pathways could block transmission while maintaining reproductive capacity. Parasitic plants and herbivores pose unique challenges through resource hijacking or direct floral damage. The synchronization of *Cuscuta* flowering with host FT protein expression suggests tissue-specific modulation of flowering regulators could disrupt parasitic associations. Similarly, engineering HIPVs through targeted overexpression of terpene synthase genes may simultaneously enhance pollinator attraction while deterring pests. Integrating these approaches into breeding programs will be crucial for developing crops that withstand biotic stresses without compromising reproductive success. Future research should prioritize functional validation of candidate genes identified from transcriptomic and GWAS studies. By bridging mechanistic insights with innovative breeding technologies, we can enhance global food security in the face of mounting environmental challenges.

## ACKNOWLEDGEMENTS

We thank Dr. Alan K. Chang for his kind effort in revising the language of the manuscript.

### Funding

This research was funded by the Zhejiang Provincial Natural Science Foundation of China (ZCLZ25C0201), and the Graduate Scientific Research Foundation of Wenzhou University (3162024003063). The funders had no role in study design, data collection and analysis, decision to publish, or preparation of the manuscript.

## Grant Disclosures

The following grant information was disclosed by the authors:

Zhejiang Provincial Natural Science Foundation of China: ZCLZ25C0201.

Graduate Scientific Research Foundation of Wenzhou University: 3162024003063.

## Competing Interests

The authors declare that they have no competing interests.

## Author Contributions

- Zhenzhen Li performed the experiments, analyzed the data, prepared figures and/or tables, authored or reviewed drafts of the article, and approved the final draft.
- Shuo Wang performed the experiments, authored or reviewed drafts of the article, and approved the final draft.
- Yike Wang performed the experiments, authored or reviewed drafts of the article, and approved the final draft.
- Hongxia Zhang performed the experiments, authored or reviewed drafts of the article, and approved the final draft.
- Lu Liu performed the experiments, authored or reviewed drafts of the article, and approved the final draft.
- Shiwen Su conceived and designed the experiments, authored or reviewed drafts of the article, and approved the final draft.
- Sue Lin conceived and designed the experiments, prepared figures and/or tables, authored or reviewed drafts of the article, and approved the final draft.

## Data Availability

No raw data was generated in our literature review.

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
