# Peer review of "The effects of biotic stress on the sexual reproduction process of flowering plants"

_PeerJ, doi:10.7717/peerj.19880_

## Round 0.1 · original submission · Major Revisions

The manuscript was reviewed by two independent experts in the field. Both the reviewers found the work interesting but raised several issues which should be addressed for further consideration. The reviewers provide detailed comments in their reviews and point out the areas where the manuscript needs to be improved. I also read the manuscript carefully and largely agree with the reviewers’ comments.

**Language Note:** The review process has identified that the English language must be improved. PeerJ can provide language editing services - please contact us at [email protected] for pricing (be sure to provide your manuscript number and title). Alternatively, you should make your own arrangements to improve the language quality and provide details in your response letter. – PeerJ Staff

Reviewer 1 ·

Basic reporting

1. Clarity and Language Quality
- The manuscript is well-written but contains grammatical issues, redundancy, and complex phrasing that need revision.

- Example (Lines 67-70):
- Issue: "Significantly impact" is vague. How do they impact reproduction?
- Revision: Specify mechanisms, e.g., "Biotic stresses alter flowering genes (FLC, FT, SOC1), disrupt floral organogenesis, and impair pollen viability."

2. Background and Literature Coverage
- The review lacks a clear rationale for why it is needed. The introduction should define:
- What gaps exist in prior reviews on abiotic vs. biotic stress?
- What novel insights this review contributes.
- The claim of no comprehensive reviews on biotic stress is inaccurate—recent studies (e.g., Nawaz et al. 2023, Begcy et al. 2024) have covered aspects of this topic. The authors should differentiate their work.

- Conflicting Findings: Some studies show accelerated flowering (Lyons et al. 2015, Kim et al. 2017), while others report a delay (Tang et al. 2013). The manuscript should contrast and explain these discrepancies.

3. Figures and Tables: Add a better resolution for both the images
- Figure 1: Lacks specific molecular interactions. Consider adding detailed pathway diagrams.
- Figure 2: Infection/feeding site depictions are too general—add spatial and temporal progression of infections in reproductive organs.
- Missing Summary Table: summarize how different biotic stressors affect pollen viability, fertilization, seed set, and floral development.

Experimental design

1. Literature Selection and Bias:

The methodology for literature selection is not rigorous or transparent. The search terms used (Lines 80-92) are broad but lack exclusion criteria.

Key concerns:
What percentage of the studies are review-based vs. original research?
Were any studies excluded due to conflicting results?
How were studies weighted for inclusion? Did the authors prioritize recent publications, high-impact journal articles, or comprehensive experimental studies?


Suggested Fix: Add a PRISMA-style flowchart or explicit section on inclusion/exclusion criteria.

Validity of the findings

1. Lack of Contrasting Perspectives
Many findings are reported without a discussion of exceptions or contradictory results.
Example: Some plant viruses accelerate flowering (e.g., Körber 2013, BYDV in maize), while others delay it (e.g., Riedell et al. 1999, BYDV in wheat). The authors do not explore why this discrepancy occurs.

Suggested Fix: Introduce a dedicated section discussing inconsistencies and the factors (e.g., plant species, virus load, climate) that may drive variation in responses.

Additional comments

Lack of Applied Context:
How can knowledge of biotic stress impacts on reproduction be applied to breeding or agronomic management? What strategies (e.g. for stress resistance) could be developed?

Suggested Fix: Add a “Future Directions and Applications” section focusing on agricultural implications.

Reviewer 2 ·

Basic reporting

This review represents a co-review, thus we are writing as "we."

The review is of broad and cross-disciplinary interest and is within the scope of the journal. The introduction section provides short but useful background on plant reproduction (the development of organs), and brief broad coverage of the range of biotic stresses that can influence plant reproduction. The last 4 sentences (starting line 70) are all novelty statements – we recommend reducing (and altering) these, as this is ~25% of the space in the introduction. We note that many of these topics have been reviewed recently in more narrow reviews. However, we think the case-study focus of this review may yet make it a useful product, see "Study Design.”

Some language and incorporation of illustrated materials could be clearer, and the source of information for the figures is not always clear, see “Additional Comments.”

Experimental design

The authors divide their review into sections based on the type of biotic agent that affects the plant (taxonomically), which we find makes logical sense.

The literature search does not appear comprehensive, and lacks detail in the methodology of the search (i.e. what combinations of search terms were included?). However, because this topic is so broad, we are not sure that a comprehensive search is desirable. A comprehensive search would likely return tens or hundreds of thousands of papers, and it would be impossible to “thoroughly read” let alone summarize all of these. We suggest the authors should either narrow their scope, or explicitly frame their review as case studies for a few examples. We find the collection of case studies in each section was potentially useful reading. Indeed, we encourage the authors to frame their review specifically around this. I.e. by saying “Rather than comprehensively reviewing the massive amount of literature in this field, our review intends leverage case studies to cover a wide range of mechanisms by which biotic agents affect sexual reproduction in plants.”

The density of citations seems appropriate, and the major sections are well organized. However, within major sections (i.e. those focused on fungi, viruses, etc) some subsections only focus on one key study (e.g. Fungi infections impede floral organ development, and Fungi infections affect seed development) while others give less detail about many more. We recommend switching to a consistent depth (number of examples) between sections, and more information about fewer examples may actually be the most useful.

Validity of the findings

The authors lay out in the introduction that they intend to produce a brief review on effects of all sources of biotic stress on all parts of the plant reproduction pathway. The fact that biotic interactions that are negative (stressful) from the perspective of the plant influence plant reproduction is indeed well supported. Though this conclusion is not novel, we do not need the supported statements to be novel to make this a valid review contribution.

The conclusion does identify unresolved questions, but brings up a surprising amount of new ideas which are not carefully linked to the rest of the manuscript content. We recommend focusing the identification of unresolved questions around the case studies, given that the content is primarily case studies. We recognize that unresolved questions focused on the case-studies might already be what the authors intended, but this wasn’t clear: if the conclusion open questions were intended to point back to the case studies, we recommend being more explicit.

Additional comments

Line 59: Repetitive wording (example: Several excellent reviews)

Line 228: Use of the figure could be better integrated into the paragraph rather than having one sentence at the end of the paragraph. This is repeated throughout the manuscript.

Line 340: Unclear of the flow of this sentence, does the following sentence only relate to the bacterial pathogen that infects walnuts?

Lines 408 and 409: The definitions of hemiparasitism and holoparasitism could be better integrated rather than in parentheses.

---

## Round 0.2 · Major Revisions

I believe the manuscript still needs a major revision. The reviewer 1 have several concerns which should be addressed before acceptance.

Reviewer 1 ·

Basic reporting

1. Sections describing molecular pathways (e.g., fungal modulation of flowering integrator genes such as FLC, FT, and GI on pages 8–9) are overly descriptive, lacking succinct analytical synthesis.

2. Figure 2 (illustrating biotic stress mechanisms), are excessively detailed and lack clear explanatory legends or annotations, which complicates interpretation.


Revising figures for conciseness, clarity, and explicit labeling would significantly enhance readability and comprehension.

Experimental design

1. The manuscript details a clear two-step database search using Web of Science and PubMed . However, criteria prioritizing recent literature and high-impact journals potentially overlook important foundational or less-cited studies. Risk of selection bias due to stated prioritization may limit comprehensive coverage of molecular and ecological mechanisms.

Recommend: Include explicit discussion in the methods regarding the rationale for literature selection criteria and acknowledge potential biases.

Validity of the findings

1. Critical analysis of experimental evidence supporting key claims (e.g., interactions between fungi such as Piriformospora indica or Pochonia chlamydosporia and floral integrators like FT, SOC1, FLC on pages 8–9; viral infection pathways affecting pollen viability and germination in cases like PNRSV and ToBRFV on pages 11–12) is limited.

2. Insufficient attention to contradictory findings or experimental limitations within the discussed studies, weakening the critical depth.


Recommendations:
1.Incorporate specific evaluations of experimental design strengths or weaknesses for pivotal studies cited (e.g., controls used, robustness of molecular techniques such as qRT-PCR or mutant analyses in cited fungal and viral studies).

2. Explicitly address and critically compare conflicting results for example, cases where pathogens either promote or inhibit flowering time (e.g., Fusarium o versus Ustilaginoidea v) to provide nuanced interpretations.

Additional comments

1. The review currently lacks clear, detailed comparative insights across biotic stress categories (fungi vs. viruses vs. bacteria vs. herbivores vs. parasitic plants.

2. Brief mention of miRNA and lncRNA roles (page 20) lacks sufficient detail or reference to specific case studies or experiments.

3. Elaborate on agricultural relevance and practical breeding strategies derived from reviewed mechanisms

4. Clearly define targeted experiments or methodologies for instance, CRISPR-based functional validation of candidate genes identified from GWAS or transcriptome studies in plant-virus interactions

Reviewer 2 ·

Basic reporting

We (we are co-reviewers) were the prior R2, and all the previous positive comments we made in this section remain. The authors have made substantial improvements across the manuscript. We commend the new framing of the rationale, and we find the new Table 1 to be very useful. In addition, the authors have sufficiently improved the written English.

However, we still find that the source of the information in the figures is not clear. For example, in Figure 1A, AGL24, MYB5, SCO1, RGA1 are shown, but never discussed in the caption or main text and therefore also have no references to support the diagrammed pathways. This is a common pattern across figures. One very quick and easy solution would be to provide citations in the figure captions, which are short and have room for this.

Experimental design

All of the previous positive comments we made in this section remain. We also find that the organization of ideas has improved.

The new text in the survey methodology section makes clear that a vast swath of literature was sampled in an unbiased way. It is rarely possible to capture every single study on a topic, even with more than sufficient effort, thus we suggest switching from “All the articles retrieved” to “Articles retrieved” on line 95, a very minor change.

Validity of the findings

All of the previous positive comments we made in this section remain. We also find the conclusion has improved sufficiently.

Additional comments

We find that all comments we made in this section were sufficiently addressed by the authors. We especially appreciate the incorporation of figure references throughout paragraph text.

---

## Round 0.3 · Minor Revisions

Thanks for revising the manuscript. I would appreciate if authors fix the minor corrections suggested by the reviewer.

Reviewer 1 ·

Basic reporting

1. The author has done a good job addressing majority of issues suggested in the previous review.
2. The quality of table & figures have significantly improved too.
3. They also now mention clearly reason behind selecting the journals to how the approach can help agriculture in future.
4. Very minor formatting suggestions: Make sure the heading on the page goes along with the text (e.g. 'Introduction' at the end of the line on page 6 could potentially be in the same as paragraph that follows, Parasitic plants and hosts interact reciprocally in flowering (pg#16), Keep Line space for subtopics consistent)

Experimental design

The selection method is clearly mentioned

Validity of the findings

No comment

---

## Round 0.4 · accepted · Accept

The current version is satisfactory and can be published in its present form.